# Ideal Injection Points for Botulinum Neurotoxin for Pectoralis Minor Syndrome: A Cadaveric Study

**DOI:** 10.3390/toxins15100603

**Published:** 2023-10-07

**Authors:** Ji-Hyun Lee, Hyung-Jin Lee, Kyu-Ho Yi, Kang-Woo Lee, Young-Chun Gil, Hee-Jin Kim

**Affiliations:** 1Department of Anatomy and Acupoint, College of Korean Medicine, Gachon University, 1342 Seongnam-daero, Seongnam-si 13120, Republic of Korea; anatomy@gachon.ac.kr; 2Department of Anatomy, Catholic Institute for Applied Anatomy, College of Medicine, The Catholic University, 222 Banpo-daero, Seoul 06591, Republic of Korea; leehj221@catholic.ac.kr; 3Division in Anatomy and Developmental Biology, Department of Oral Biology, Human Identification Research Institute, BK21 FOUR Project, College of Dentistry, Yonsei University, 50-1 Yonsei-ro, Seoul 03722, Republic of Korea; kyuho90@daum.net (K.-H.Y.); leekw@yuhs.ac (K.-W.L.); 4Department of Anatomy, College of Medicine, Chungbuk National University, 1 Chungdae-ro, Cheongju-si 28644, Republic of Korea; gilyc@chungbuk.ac.kr

**Keywords:** pectoralis minor muscle, Pectoralis Minor Syndrome, botulinum neurotoxin, Sihler staining, thoracic outlet syndrome

## Abstract

Pectoralis Minor Syndrome (PMS) causes significant discomfort due to the compression of the neurovascular bundle within the retropectoralis minor space. Botulinum neurotoxin (BoNT) injections have emerged as a potential treatment method; however, their effectiveness depends on accurately locating the injection site. In this study, we aimed to identify optimal BoNT injection sites for PMS treatment. We used twenty-nine embalmed and eight non-embalmed human cadavers to determine the origin and intramuscular arborization of the pectoralis minor muscle (Pm) via manual dissection and Sihler’s nerve staining techniques. Our findings showed the Pm’s origin near an oblique line through the suprasternal notch, with most neural arborization within the proximal three-fourths of the Pm. Blind dye injections validated these results, effectively targeting the primary neural arborized area of the Pm at the oblique line’s intersection with the second and third ribs. We propose BoNT injections at the arborized region within the Pm’s proximal three-fourths, or the C region, for PMS treatment. These findings guide clinicians towards safer, more effective BoNT injections.

## 1. Introduction

Compression of the brachial plexus and surrounding blood vessels can affect sensation and movement in the neck, chest, shoulders, arms, and hands [1,2,3,4,5]. This neurovascular bundle passes through the scalene triangle above the clavicle, the costoclavicular space directly below the clavicle, and the pectoralis minor space below the clavicle. Neurovascular compression mainly occurs within these three anatomical compartments, and can be classified into thoracic outlet syndrome (TOS) and pectoralis minor syndrome (PMS) according to the location of compression with respect to the clavicle [3,6,7].

TOS and PMS have similar symptoms, and are often concurrent; therefore, history and physical examination are similar for both. Their treatment options, ranging from conservative treatment to surgical management, are similar as well [3]; however, the target muscles are different, with the anterior scalene muscle being targeted for the treatment of TOS and the pectoralis minor (Pm) muscle for treatment of PMS [7,8]. PMS is caused by compression of the neurovascular bundle within the retropectoralis minor space. It has various causes, including traumatic events, anatomical deformations, and atrophy of the flexor muscles including of the pectoralis minor due to hemiplegia and compression as a result of stiffness. The characteristic physical signs of PMS are pain and tenderness of the Pm muscle under the clavicle and armpit, and weakness, cyanosis, edema, and paresthesia may develop as well [3,5]. A tightened and compressed pectoralis minor muscle is predicted to contribute to the progression of soft tissue contractures, leading to postural deformities such as rounded shoulders. The reduction of muscle tightness prevents disease progression, overcomes functional impairment, and may contribute to pain relief and restoration of ideal posture [8,9]. For PMS, conservative treatments such as stretching and surgical treatments such as tenotomy are available, and the use of nonsurgical treatments, including corticosteroid injections, anesthetics, and botulinum neurotoxins (BoNT) has increased [3,6,7,10,11,12].

BoNT injections may be tried before surgery or for PMS that is not relieved by surgery and other treatments. Relative to surgical treatment, such injections are inexpensive, practical, and above all, safe [5,13]. BoNT injections into the muscle adjacent to the brachial plexus may induce muscle relaxation and reduce nerve compression; therefore, they may provide symptomatic relief. Several previous studies have demonstrated therapeutic outcomes for PMS through administration of BoNT [5,6,7,14]. When administering BoNT, both the dosage and the precise injection site are critically important [15,16,17]. Additionally, a differential diagnosis between TOS and PMS can be facilitated by performing a diagnostic nerve block of the medial and lateral pectoral nerve before treating PMS [3,18,19]. For this procedure, understanding the location of the Pm muscle is crucial. To achieve the most effective therapeutic outcome with BoNT, it is essential to target the intramuscular neural arborization [20,21]. In particular, when injecting into the Pm muscle, injection at an inappropriate site can lead to a range of side effects due to its anatomical positioning, from unwanted paralysis of the surrounding muscles to more severe complications such as pneumothorax [3].

In order to achieve precise injections into the Pm muscle and mitigate the risk of pneumothorax arising from the inherent complexities associated with thoracic region injections, the employment of ultrasound-guided injections has been persistently advocated in the literature [6,22]. Nonetheless, in certain clinical settings constrained by budgetary restrictions it may be the case that access to the requisite ultrasound devices is lacking. Furthermore, seasoned clinicians may continue to employ blind injection techniques in the absence of ultrasound guidance [3].

Extant studies have primarily focused on the relationship between the origins of the Pm in the ribs, the morphology of the insertion, or other variations. Therefore, there have been few detailed studies on the location of the muscles, and existing studies have not elucidated the intramuscular arborized pattern that can be inferred from the motor endplate of the muscle. This is probably because dissection limits the tracking of microscopic innervation due to the risk of mechanical damage to the nerves [23].

This study aims to provide guidelines for more effective and safer BoNT injections based on anatomical findings, as follows: (1) identification of the origin of the Pm based on surface landmarks; (2) identification of intramuscular arborization of the Pm using Sihler’s staining method; and (3) validation of the proposed blind injection using fresh cadavers based on the results of this study. Consequently, based on the results of this study, we aim to address blind injection methods while briefly discussing the importance of ultrasound-guided injections.

## 2. Results

### 2.1. Locations of the Pm Muscle: Origin and Nerve Entry Points


The Pm muscle had various origins. It originates from the regions of the third to fifth ribs, second rib to the fourth intercostal fascia, and second to fourth ribs in 30%, 15%, and 12% of cases, respectively (Table 1). The Pm muscle mainly covers the second and third ribs on the line passing the medial one-third of the clavicle at a 45-degree angle (CL) (Table 2).

The average distance from the line passing through the suprasternal notch at a 45-degree angle (SL) to the medial border of the Pm muscle at the third, fourth, and fifth ribs was 23.10 ± 9.09 mm, 8.94 ± 5.48 mm, and 5.29 ± 3.11 mm, respectively (Figure 1). The origins of the Pm muscle at the third and fourth ribs were located lateral to the SL in 97% and 84% of cases, respectively. For the cases where the Pm muscle originated from the fifth rib, the origin was lateral to SL in 57% of cases. The average width of the Pm at the CL was 68.18 ± 8.96 mm. The average lengths of the superior and inferior borders of the Pm muscle were 94.51 ± 13.84 mm and 136.05 ± 13.36 mm, respectively. The average length of the clavicle was 145.35 ± 13.86 mm.

The nerve entry points had several locations in each muscle, and were mostly distributed in the B area. The nerve entry points were observed at the B1 area for 90% of the cases, B2 area for 67% of the cases, and B3 area for 90% of the cases (Figure 2A).

### 2.2. Intramuscular Arborization Patterns of the Pm Muscle

The intramuscular arborization pattern was observed at several locations in each muscle, predominantly in the C area. The arborization area was observed in the C1 area for 73% of the cases, C2 and C3 area for 100% of the cases (Figure 2B).

### 2.3. Blind Dye Injection of the Pm Muscle of Fresh Cadavers

For 14 out of 15 injections (93%), it was confirmed that the dye was injected into the Pm muscle (Figure 3B). In one case, only one of the two injections (upper portion) was successful. Therefore, we categorized this case as a failure. There was no specific variation of the Pm muscle in that specimen. Sihler’s staining confirmed that the injections were near the neural arborized area (Figure 4).

## 3. Discussion

This study revealed new anatomical information about the Pm muscle. As a landmark, the length of the clavicle was measured to ascertain the location of the Pm muscle; the result was similar to that of a previous study on the clavicle [24]. However, because previous studies on the Pm muscle mainly focused on its insertion and variation, there were only a few studies for comparison. A previous study reported that the Pm muscle mainly originated from the second to fifth ribs and the third to fifth ribs in 42% and 29% of cases, respectively [25]. In this study, the Pm muscle originated from the third to fifth ribs in most cases, accounting for 30%, which was similar to the results in [25]. However, the Pm muscle originated from the second to fifth ribs for only 6% of the cases in this study, which is different from the result of the previous study, where this origin was the most common [25]. The reason for this difference is that this study involved a different racial population from that of the previous study, including cases with origins from the fascias of intercostal muscles, and not only the ribs. Additionally, the number of samples in this study is considered insufficient to classify the shape.

According to previous studies, the Pm muscle has been targeted using blind needle injection techniques. The landmark for this approach, as described in one study, is situated 4–6 cm below the clavicle, and is aligned with the most tender spot of the Pm muscle [3]. Another study has recommended the pincer palpation technique as a method to stimulate the muscle [26]. In contrast, most studies on the treatment and diagnosis of PMS using botulinum toxin have utilized ultrasound-guided injection methods [7,14,27]. This prevalent choice may be due to the limitations inherent in the aforementioned blind injection techniques, such as complications and a lack of detailed guidance. However, detailed injection methodologies are absent in these studies.

The results of this study provide several important insights into finding the ideal injection site for the Pm muscle. First, at the levels of the fourth and fifth ribs, the Pm muscle was located near the SL. From this, we speculate that the origin of the Pm muscle can be easily inferred using the SL, which is a landmark that can be easily identified on the surface. Second, the intramuscular nerve ending of the Pm muscle was mainly arborized in area C. Therefore, BoNTs should be injected into that area. The muscle volume reduction effect is greater when BoNTs are injected into the arborized area relatively close to the motor endplate instead of distally [28,29,30]. Third, the clavicle and rib were used to suggest an appropriate injection point. The CL was used because it was an easier landmark, and based on the results of this study, the CL passes near the C area of the Pm muscle. We were able to reconfirm this using ultrasonography images of living humans (Figure 5) and by dissection and Sihler’s staining in cadavers after injection. Lastly, the entry points of the Pm muscle were mainly located within area B. In previous studies, the risk of nerve damage caused by needles after injections into the nerve entry point has been mentioned [31,32]. Therefore, it is necessary to be cautious in order to prevent injecting into area B. Instead, we propose area C of the Pm muscle in order to minimize direct nerve damage caused by incorrect injections and reduce muscle volume more effectively.

In addition, we propose injecting over the rib as a safe technique even for blind injection to prevent pneumothorax. During blind injections into cadavers, we observed no instances of pneumothorax. The reason is clear, as the needle was placed above the ribs; however, injecting into the rib does not always ensure optimal efficacy, as demonstrated by one failed blind injection and observations from ultrasound. In certain situations, even a minor procedural misstep can result in pneumothorax. Therefore, in order to ensure the safest injection we strongly recommend the use of the ultrasound-guided injection. Ultrasound-guided Pm injection is likely to be safer and more effective (Figure 6). Above all, it is difficult to accurately inject only the Pm, and there is a risk of injecting into the adjacent muscles or intermuscular septal area during blind injections. The results of this study propose a more detailed injection protocol, ensuring that anyone can easily target the Pm muscle. Furthermore, these results can enhance the safety of ultrasound-guided techniques.

There are several limitations of this study that should be noted. Because the study was conducted on cadavers, we cannot confirm whether our proposal will be effective in actual PMS patients. We did not evaluate the relationship between the amount of subcutaneous fat in the cadavers and potential changes in muscle position. Additionally, we did not assess how the amount of subcutaneous tissue might influence the accuracy of blind injections. To overcome these limitations, more extensive research on living subjects should follow.

## 4. Conclusions

This study identified ideal injection points for the Pm muscle to enhance the effect of BoNTs with a minimal dose. We propose BoNT injections at multiple locations (two or more) with low doses to reduce side effects. The BoNTs should be directly injected into the neural arborized area of the Pm muscle, particularly in the C region (3/4 region). Moreover, we recommend using ultrasound guidance for Pm injections, both for accuracy and to prevent pneumothorax.

The results of this study propose a more detailed injection protocol, ensuring that anyone can easily target the Pm muscle. Additionally, these findings can potentially enhance the safety of ultrasound-guided techniques. The insights from this study may be valuable for other chest surgeries and procedures other than BoNT injections.

## 5. Materials and Methods

This study used 48 Pm muscles obtained from 37 cadavers (29 embalmed and 8 non-embalmed cadavers); these cadavers included 23 males and 16 females with a mean age of 81.1 years (range, 63–98 years). Only cadavers without surgery, trauma, or deformities in the chest region were used. This study was performed in accordance with the principles set forth in the Declaration of Helsinki. The cadavers were legally donated to the Surgical Dissection Education Center, Yonsei University College of Medicine, and appropriate consent was obtained for dissection and research (approval number: YSAEC 22-008).

### 5.1. Measurements of the Location of the Pm

#### 5.1.1. Identification of the Location of the Origin of the Pectoralis Minor Muscle Based on the Surface Landmarks

To identify the relationships between the location of the Pm muscle and the surface landmark, two reference lines were established: the line passing through the suprasternal notch at a 45-degree angle (SL), and the line passing the medial one-third of the clavicle at a 45-degree angle (CL) in the anatomical position. In all, 33 Pm muscles from 29 embalmed cadavers were used. The skin, subcutaneous tissue, and pectoralis major muscle were carefully dissected to identify the Pm muscle. The shape of the Pm muscle was confirmed and determined based on the following (Figure 7): the origin of the Pm muscle, number of ribs overlapped by the Pm muscle at CL, length of the clavicle, length of the superior and inferior borders of the Pm muscle, width of the Pm muscle at CL, and distance from the SL to the origin (muscle belly) of the Pm muscle within the region of the third to fifth ribs.

#### 5.1.2. Nerve Entry Point of the Pm Muscle

After measurements, the 21 sides of the Pm muscle were harvested to examine the nerve entry points and intramuscular arborization pattern. The procured Pm muscle was split into four transversely (A to D) and three vertically (1 to 3) along the border, thereby dividing it into 12 regions: A1–3, B1–3, C1–3, and D1–3. Due to the oblique orientation of the Pm muscle, we defined the long axis (from the origin to the insertion of the muscle) as the transverse axis and the short axis (the length at the origin or insertion) as the vertical axis. Thereafter, the location of the entry point was marked (Figure 8, top).

### 5.2. Intramuscular Arborization of the Pm Muscle

In order to identify the neural arborization pattern of the Pm muscle, we utilized the whole-mount staining technique, called “Sihler’s staining,” which essentially shows the intramuscular nerve distributions without inciting nerve damage. This method has seven steps and was used with modifications in this study. After staining, the Pm muscle was divided into 12 regions and the arborization pattern was analyzed (Figure 8, bottom).

#### Modified Sihler’s Staining

(1) Fixation step: the procured Pm was fixed for three days in 10% unneutralized formaldehyde. (2) Maceration and depigmentation: the fixed samples were macerated and depigmented for two weeks in 3% aqueous potassium hydroxide solution (adding 1 mL of 3% hydrogen peroxide to 1000 mL). (3) Decalcification and whitening step: the macerated samples were decalcificated and whitened in a solution called “Sihler I solution,” a mixture of 10% glacial acetic acid and 10% glycerin in distilled water, for three days. (4) Staining step: after the samples had been decalcified, they were stained in a solution called “Sihler II solution”, a mixture of 10% Ehrlich’s hematoxylin and 10% glycerin in distilled water, for one day. (5) Destaining step: the stained samples were again destained with Sihler I solution for three hours. (6) Neutralization and blueing step: the destained samples were neutralized in running tap water for a hour, then the sample was emerged in 0.05% lithium carbonate for a hour to blue the nerve fibers. (7) Clearing step: the neutralized samples were ultimately cleaned and transparency in 99% formamide; in the case of pm from fresh cadaver, it was cleaned in methyl salicylate.

### 5.3. Cadaveric Evaluation of Blind Injection of the Pm Muscle

In this step, blind injections were administered to 15 sides of the Pm muscles from 8 fresh cadavers. Based on the results of Sihler’s staining and previous experience, we used the clavicle as a landmark and the ribs to find a point for a more efficient injection. Approximately one millimeter of dye was injected vertically into the skin above the second and third ribs intersecting the CL (Figure 3A). At this time, we used a 3 cc syringe and a 23 gauge 25 mm needle to inject the dye. The skin and pectoralis major muscle were carefully removed after the injection, and it was confirmed that the dye was injected within the Pm muscle (Figure 3B). After that, the three Pm muscles were harvested, and Sihler’s staining was used to confirm the adequacy of the injection.

## Figures and Tables

**Figure 1 toxins-15-00603-f001:**
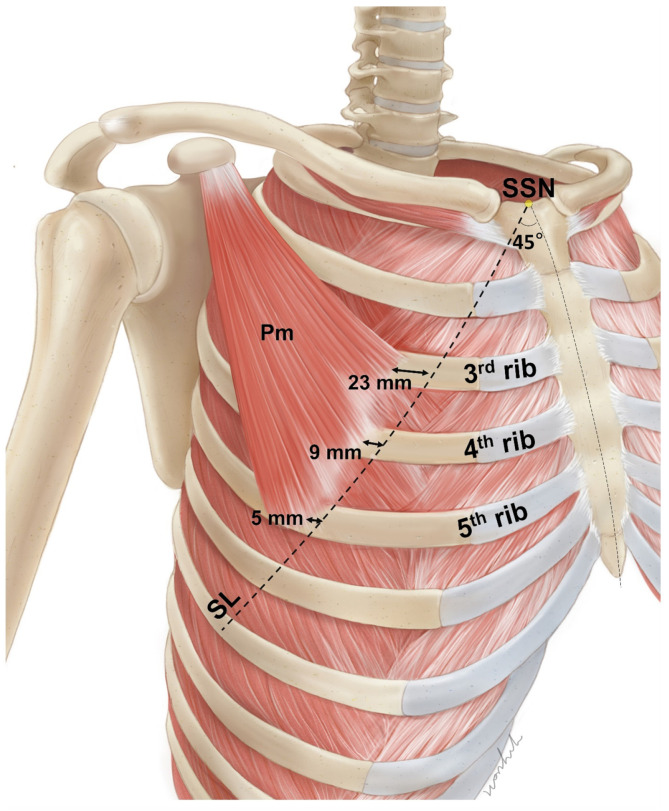
The average distance from the line passing the suprasternal notch (SSN) at a 45-degree angle (SL) to the origin of pectoralis minor (Pm). At the third, fourth, and fifth ribs, the origins of the Pm muscle are located near the SL.

**Figure 2 toxins-15-00603-f002:**
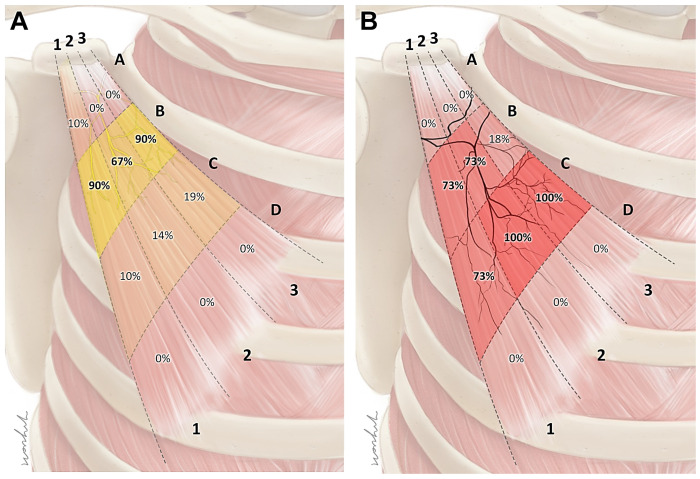
The nerve entry point pattern of the pectoralis minor muscle (**A**). The nerve entry points were mostly distributed in the B area (indicated by yellow squares). The intramuscular arborization pattern of the pectoralis minor muscle (**B**). The intramuscular arborization area was predominantly in the C area. The intensity of the color in these figures indicates the distribution ratio.

**Figure 3 toxins-15-00603-f003:**
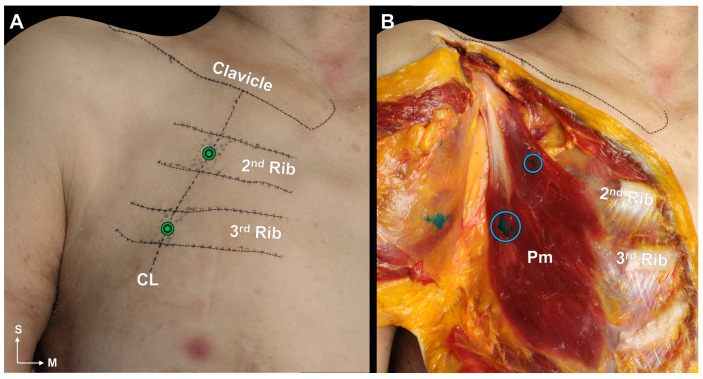
Blind injection method using the landmark (**A**). The dyes were injected above the second and third ribs intersecting the 45-degree line crossing the medial one-third of the clavicle (CL). The green circle indicates the injection points. Results of blind injection (**B**). The dyes (indicated by the blue circle) were well observed in the pectoralis minor (Pm) muscle. S, superior; M, medial.

**Figure 4 toxins-15-00603-f004:**
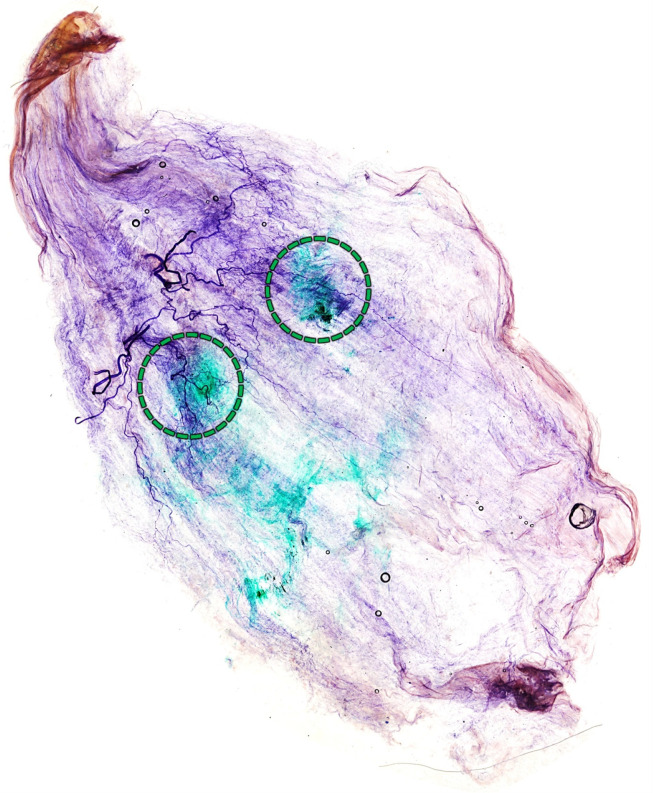
Result of Shiler’s staining of the pectoralis minor muscle of the blind dye-injected fresh cadaver. The green dye is located near the neural arborized area (indicated by a green circle). The dye in the distal part spread during the Sihler’s staining process.

**Figure 5 toxins-15-00603-f005:**
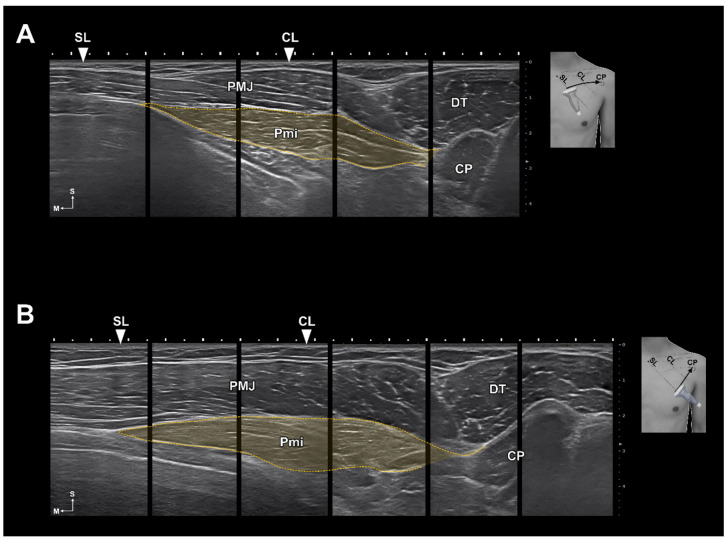
Ultrasonography images of the line passing the suprasternal notch at a 45-degree angle (SL) to the coracoid process (CP). The US images were scanned along the line connecting the CP and second rib (**A**) intersecting with the line at the medial one-third of the clavicle at a 45-degree angle (CL) and third rib (**B**). In both US images, the CL is located near the middle of the pectoralis minor muscle (Pmi, indicated by an orange dot line). In B, the medial end of the origin of the Pm muscle is located near the SL. PMJ, pectoralis major; DT, deltoid muscle; S, superior; M, medial.

**Figure 6 toxins-15-00603-f006:**
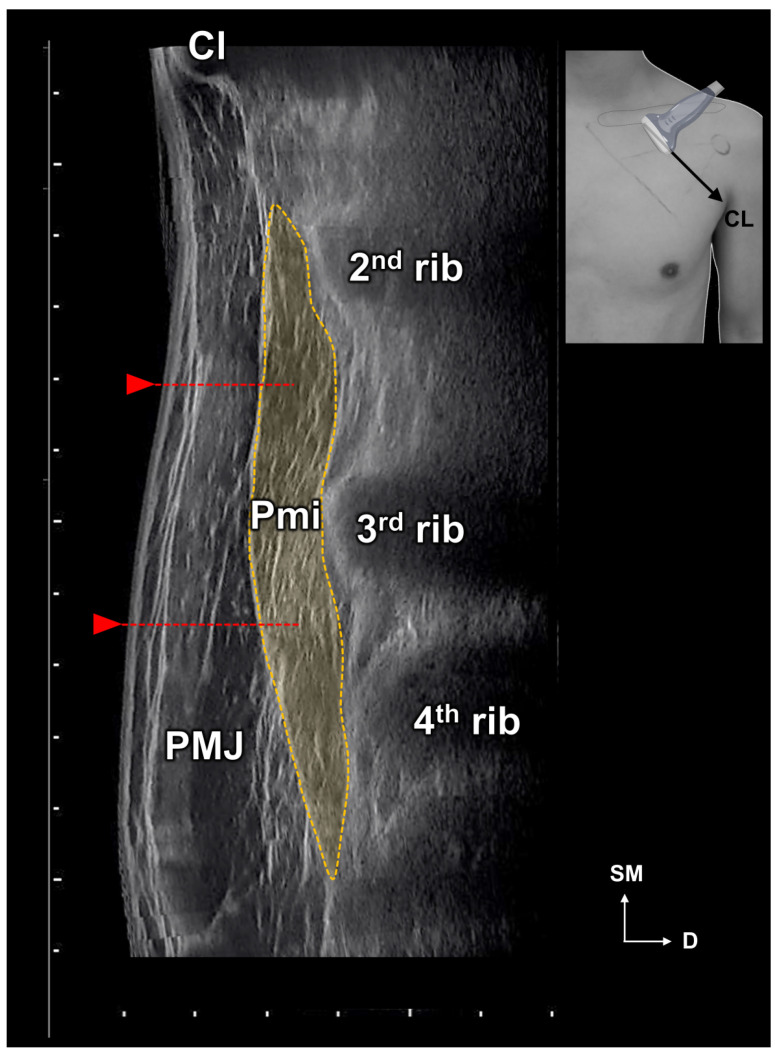
Ultrasonography images along the 45-degree line crossing the medial one-third of the clavicle (CL). In this individual, the pectoralis minor muscle (Pmi) spans the second to fourth ribs. Therefore, it is considered to inject above the second and third intercostal space (indicated by red arrows) for the most efficient and safe results. Caution should be exercised to prevent pneumothorax. PMJ, pectoralis major muscle; SM, Superomedial; D, deep.

**Figure 7 toxins-15-00603-f007:**
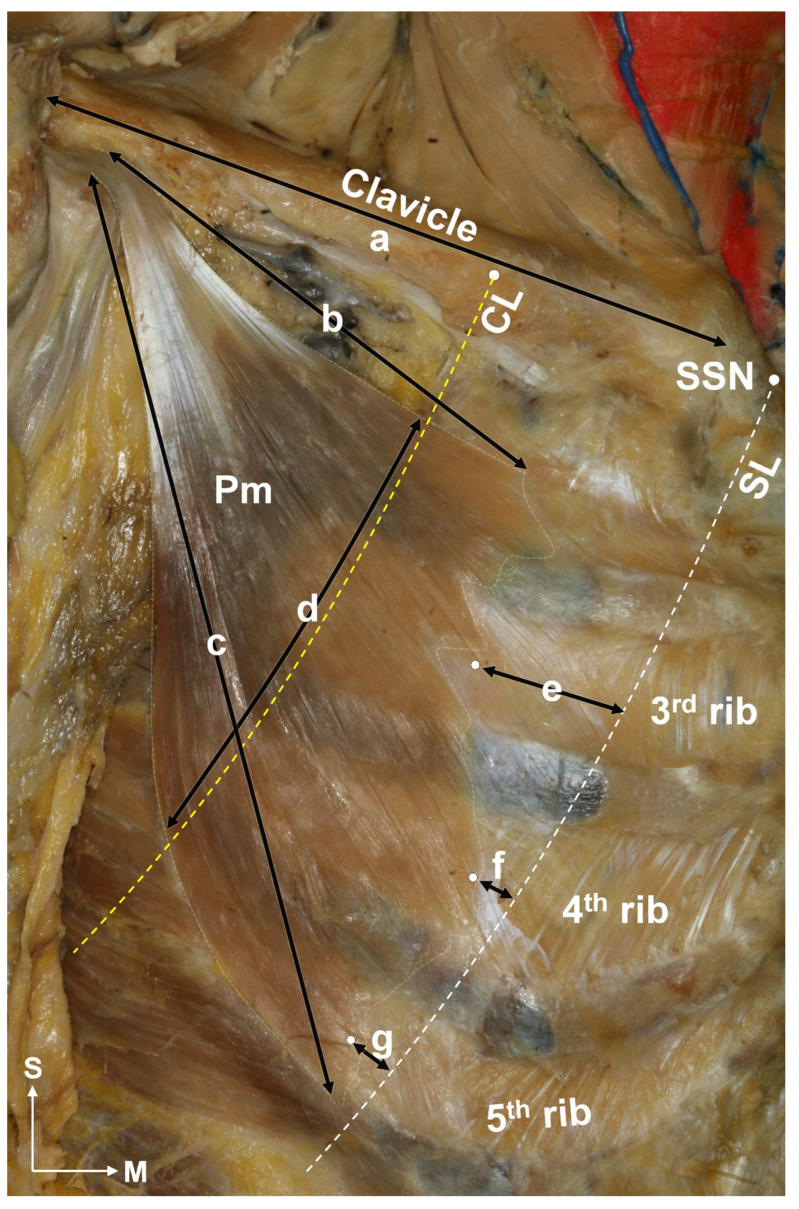
Pectoralis minor muscle measurements. The length of the clavicle (a), length of the Pm muscle (b, c), and width of the Pm muscle at the line passing the medial one-third of the clavicle at a 45-degree angle (CL, indicated by yellow dot line) (d) were measured. The distance from the line passing through the suprasternal notch (SSN indicated by white dot line) at a 45-degree angle (SL) to the origin of the Pm muscle was measured at the level of the third to fifth ribs (e to g). S, superior; M, medial.

**Figure 8 toxins-15-00603-f008:**
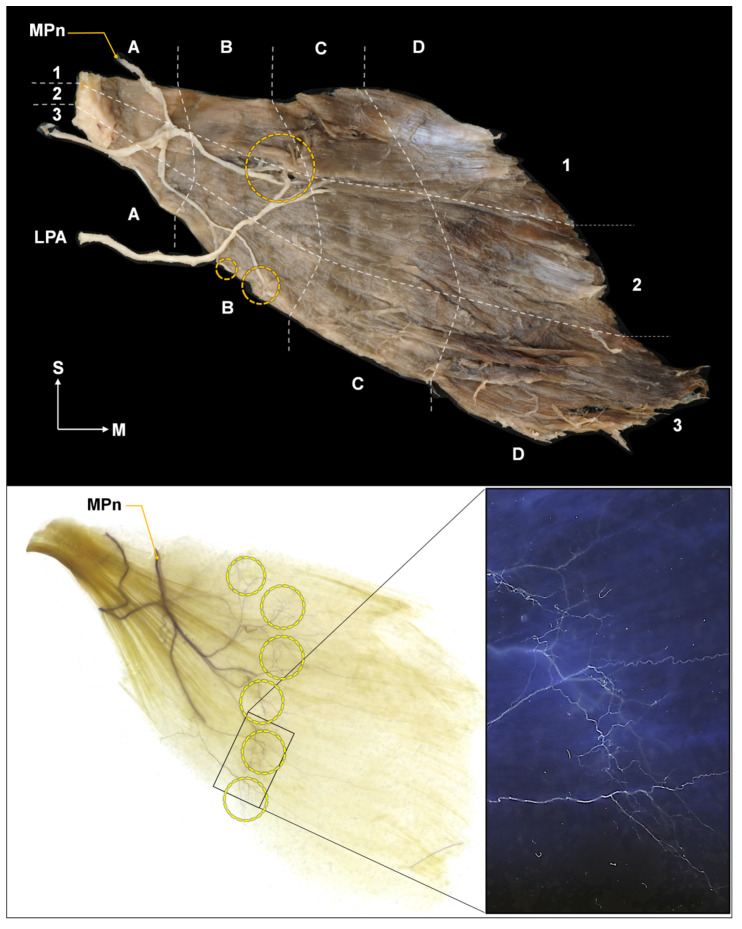
Identification of the nerve entry point of the pectoralis minor muscle (**top**). The Pm muscle was divided equally according to the shape of the muscle into four horizontally (A to D) and three vertically (1 to 3), and the medial pectoral nerve (MPn) was traced to identify the entry point (indicated by the orange circle). Result of Sihler’s staining (**bottom**). Tiny nerve branches of the MPn form intramuscular neural arborization, indicated by a yellow circle. S, superior; M, medial; LPa, lateral pectoral artery.

**Table 1 toxins-15-00603-t001:** The location of the origin of the pectoralis minor muscle from the ribs.

Rib Number *	1.5–3.5	2–4	2–4.5	2–5	2.5–4	2.5–4.5	2.5–5	3–5	3–5.5	3–6
Cases	1	4	5	2	2	3	3	10	2	1
Percentage	3%	12%	15%	6%	6%	9%	9%	30%	6%	3%

* 0.5 means origin from the intercostal space.

**Table 2 toxins-15-00603-t002:** Ribs covered by the pectoralis minor muscle along the CL line.

Rib Number	2nd and 3rd Rib	2nd, 3rd, and 4th Rib	3rd, 4th, and 5th Rib
Cases	10/33	22/33	1/33
Percentage	30%	67%	3%

## Data Availability

The datasets used and/or analyzed during the current study are available from the corresponding author upon reasonable request.

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
