# Peer review of "Ideal Injection Points for Botulinum Neurotoxin for Pectoralis Minor Syndrome: A Cadaveric Study"

_toxins, 2023, doi:10.3390/toxins15100603_

Round 1
Reviewer 1 Report
This is an extremely well written article that provides beneficial anatomical information that will educated and facilitate injections of botulinum toxin into the pectoralis minor muscle.
Author Response
Dear Reviewer,
We are profoundly grateful for your thoughtful and constructive review of our manuscript. Your insights are invaluable, and we genuinely appreciate the time and expertise you have invested in evaluating our work.
In response to the feedback from you and other esteemed reviewers, we are in the process of making comprehensive revisions to our manuscript. Once these are complete, we intend to resubmit for your further consideration.
Our team is steadfast in our commitment to advancing the field through rigorous research. We assure you of our continued efforts to contribute meaningful and impactful studies to the scientific community.
With deepest gratitude and respect,
Reviewer 2 Report
Thank you for the opportunity to review this article concerning the identification of anatomical markers for the blind identification of the pectoralis minor muscle in the context of pectoralis minor syndrome. The article appears as interesting even if the thoracic outlet syndrome resulting from hypertrophy of the pectoralis minor muscle represents a facet of the possible interest of readers for this article. The pectoralis minor is not the target only in this syndrome but also, for example, in the patient suffering from spasticity with adducted, anteposed and painful shoulder and this article could also be interesting in this context. In this way, however, it should be rethought to make it more generalist and focused on the muscle and not on the condition/illness. I think this is feasible and might broaden the audience of possible readers.
Below are listed minor and major hints:
Major
- Introduction: about BoNT-A lines 51-54 page 2. Specify more about BoNT-A because 23-50U are not informative because the Unit is not interchangeable between the different specialities of BoNT-A. Specify also the dilution that affects the diffusion of the drug injected. Furthermore, the involvement of the adjacent muscles and the autoantibody production is not related primarily to the dose but to other issues such as for the former the technique of injection and volume and for the latter the state of the patients e.g. fever by recent infections.
- Materials and methods chapter 2.3: this paragraph suffers from some defect. First of all the measure of the depth of the injection of 25mm cannot fail to consider that these are cadaveric preparations and subjects with an average age of 80 years. In these conditions, the subcutaneous tissue is less represented. Moreover, according to the pathology for which the article was produced, the pathology of the thoracic outlet generally involves younger subjects and therefore the cadaveric tissue used does not reflect the possibly affected patient in terms of homogeneity of conditions. Finally due to the diffusion of the ultrasound seems almost anachronistic to define blind methods for the injection of the pectoralis minor also considering the risks of associated pneumothorax.
-
Minor
- Abstract: use the same format for the numbers e.g. "29 ambalmed and eight non-embalmed".
- Before treating the PM syndrome also diagnostic nerve block of the pectoralis minor nerve can be performed before the treatment with BoNT-A or other treatment if the cause is the contracture of the muscle and this cannot be ignored.
- Discussion line 200 page 8: this concept is already known but there are articles that investigated the end plate as a valuable target of BoNT-A injection that evaluate many muscles and not only the psoas as in the one cited, reference 20.
- Figure 8 appears as not right because if the US probe was placed as in the picture the US image is not the same. Appears as rotated.
Author Response
We wish to express our profound gratitude for your comprehensive review and the detailed suggestions you provided for our manuscript. Your feedback has been instrumental in identifying areas of improvement, and we genuinely appreciate the depth of expertise you brought to the evaluation process.
In response to your feedback, we have made the necessary revisions to our manuscript. We are submitted the revised content along with a PDF version of the updated manuscript for your further consideration. Please see the attachment
Your guidance has significantly enhanced the quality and rigor of our work, and we are deeply thankful for your contribution to refining our research.
With sincere appreciation and respect,

Reviewer 3 Report
The authors have studied an interesting and clinically relevant anatomical topic. The manuscript is written nicely and the results presentation was sufficient. There are a few points that the authors need to consider to improve the reader's interest in the subject.
1. The authors might consider discussing the current practice of botulinum PMi injection to showcase the importance of their study. US-guided injections are already in use and it will be nice to elaborate on the reported technique. This can be used to improve the discussion accordingly.
2. Details about how the muscle was divided (section 2.1.2) are needed, and attention also to which lines were vertical and longitudinal. This is necessary as the reported origins of the Pm vary and thus we need to know the surface relevance of these points to the injection technique.
3. The discussion's first two paragraphs are mainly a repetition of the introduction. It is advisable to concentrate on what this study added to the reader's knowledge and correlate that to the available knowledge as exactly done in the following paragraphs.
4. The usage of US in studying the muscle appeared only in the discussion, which could have added more applicability to the study if it was reported in the methods/results.
5. Minor points:
- Line 135 & 136: lateral to Cl, I believe that the origin of Pm is medial to the CL. kindly revise.
- Title of table 1: are you referring to the origin of the muscle here. it was written (insertion).
- line 150: can the authors comment of the failed injection if it was due to anatomical variation in the muscle.
- Line 151: Figure 4.B, I think it should be Figure 3.B
-Lines 185, 202: previous study/ies. Kindly add references.
- the conclusion that the authors recommend US in the injection does not seem to be concluded from the presented study as it is the current practice. It would be better if they emphasize on modifying/considering the new findings to guide the US injection.
Thank you
Author Response
We wish to express our profound gratitude for your comprehensive review and the detailed suggestions you provided for our manuscript. Your feedback has been instrumental in identifying areas of improvement, and we genuinely appreciate the depth of expertise you brought to the evaluation process.
In response to your feedback, we have made the necessary revisions to our manuscript. We are submitted the revised content along with a PDF version of the updated manuscript for your further consideration. Please see the attachment.
Your guidance has significantly enhanced the quality and rigor of our work, and we are deeply thankful for your contribution to refining our research.
With sincere appreciation and respect,

Round 2
Reviewer 2 Report
Dear Authors I appreciate all of your comments and corrections that follow the suggestions received. All major hints are excellently resolved or added as limits where appropriate.
Only one minor edit appears appropriate: the target of the diagnostic nerve block is not the muscle but the respective nerve (page 2 lines 52-54) so might be preferable to state simply as follows: " ...can be facilitated by performing a diagnostic nerve block of the medial and lateral pectoral nerve".
(Winston P et al. Recommendations for Ultrasound Guidance for Diagnostic Nerve Blocks for Spasticity. What are the Advantages? Archives of Physical Medicine and Rehabilitation 2023;
Alain P. Yelnik French clinical guidelines for peripheral motor nerve blocks in a PRM setting Annals of Physical and Rehabilitation Medicine 2019).
Author Response
We would like to extend our deepest gratitude for your meticulous review and invaluable suggestions, which have significantly contributed to the enhancement of our manuscript. In accordance with the insights provided by the reviewer, we have undertaken the following revisions:
Before (page 2 lines 52-54)
Additionally, before treating PMS, a differential diagnosis between TOS and PMS can be facilitated by injecting Lidocaine into the Pm muscle...
→ After
Additionally, a differential diagnosis between TOS and PMS can be facilitated by performing a diagnostic nerve block of the medial and lateral pectoral nerve before treating PMS [3,18,19].
18. Yelnik, A.P.; Hentzen, C.; Cuvillon, P.; Allart, E.; Bonan, IV.; Boyer, F.C.; Coroian, F.; Genet, F.; Honore, T.; Jousse, M.; Fletcher, D.; Velly, L.; Laffont. I.; SOFMER group; SFAR group; Viel, E. French clinical guidelines for peripheral motor nerve blocks in a PRM setting. Ann Phys Rehabil Med. 2019, 62, 252-264, doi: 10.1016/j.rehab.2019.06.001. Epub
19. Winston, P.; Reebye, R.; Picelli, A.; David, R.; Boissonnault, E. Recommendations for Ultrasound Guidance for Diagnostic Nerve Blocks for Spasticity. What Are the Benefits? Arch Phys Med Rehabil. 2023, 104, 1539-1548, doi: 10.1016/j.apmr.2023.01.011. Epub